

# Genetic variation of Nigerian cattle inferred from maternal and paternal genetic markers

David H. Mauki[1,2,3,*], Adeniyi C. Adeola[1,2,*], Said I. Ng'ang'a[1,2,3], Abdulfatai Tijjani[4], Ibikunle Mark Akanbi[5], Oscar J. Sanke[6], Abdussamad M. Abdussamad[7], Sunday C. Olaogun[8], Jebi Ibrahim[9], Philip M. Dawuda[10], Godwin F. Mangbon[11], Paul S. Gwakisa[12], Ting-Ting Yin[1], Min-Sheng Peng[1,2,3] and Ya-Ping Zhang[1,2,3,13,14]

[1] State Key Laboratory of Genetic Resources and Evolution, Kunming Institute of Zoology, Chinese Academy of Sciences, Kunming, Yunnan, China
[2] Chinese Academy of Sciences, Sino-Africa Joint Research Center, Kunming, Yunnan, China
[3] University of Academy of Sciences, Kunming College of Life Science, Kunming, Yunnan, China
[4] School of Life Sciences, University of Nottingham, Nottingham, UK
[5] Ministry of Agriculture and Rural Development, Secretariat, Ibadan, Oyo, Nigeria
[6] Taraba State Ministry of Agriculture and Natural Resources, Jalingo, Taraba, Nigeria
[7] Department of Animal Science, Faculty of Agriculture, Bayero University, Kano, Kano, Nigeria
[8] Department of Veterinary Medicine, University of Ibadan, Ibadan, Oyo, Nigeria
[9] College of veterinary medicine, department of theriogenology, University of agriculture, Makurdi, Makurdi, Benue, Nigeria
[10] Department of Veterinary Surgery and Theriogenology, College of Veterinary Medicine, University of Agriculture Makurdi, Makurdi, Benue, Nigeria
[11] Division of Veterinary Office, Serti, Taraba, Nigeria
[12] Department of Microbiology, Parasitology and Biotechnology/ Genome Science Center, Sokoine University of Agriculture, Morogoro, Tanzania
[13] State Key Laboratory for Conservation and Utilization of Bio-Resource in Yunnan, School of Life Sciences, Yunnan University, Kunming, Yunnan, China
[14] Center for Excellence in Animal Evolution and Genetics, Chinese Academy of Sciences, Kunming, Yunnan, China
* These authors contributed equally to this work.

Corresponding author
Ya-Ping Zhang,
zhangyp@mail.kiz.ac.cn

## ABSTRACT

The African cattle provide unique genetic resources shaped up by both diverse tropical environmental conditions and human activities, the assessment of their genetic diversity will shade light on the mechanism of their remarkable adaptive capacities. We therefore analyzed the genetic diversity of cattle samples from Nigeria using both maternal and paternal DNA markers. Nigerian cattle can be assigned to 80 haplotypes based on the mitochondrial DNA (mtDNA) D-loop sequences and haplotype diversity was 0.985 + 0.005. The network showed two major matrilineal clustering: the dominant cluster constituting the Nigerian cattle together with other African cattle while the other clustered Eurasian cattle. Paternal analysis indicates only zebu haplogroup in Nigerian cattle with high genetic diversity 1.000 ± 0.016 compared to other cattle. There was no signal of maternal genetic structure in Nigerian cattle population, which may suggest an extensive genetic intermixing within the country. The absence of *Bos indicus* maternal signal in Nigerian cattle is attributable to vulnerability bottleneck of mtDNA lineages and concordance with the view of male zebu genetic introgression in African cattle.

Our study shades light on the current genetic diversity in Nigerian cattle and population history in West Africa.

# INTRODUCTION

The modern domestic cattle were initially domesticated about 10,000 years ago from two putative domestication centers, the Near East for *Bos taurus* and the Indian Sub-continent for *B. indicus* (*Loftus et al., 1994a*). In Africa, the modern domestic cattle were probably introduced at different times with *B. taurus* circa 7,000–4,000 years BP and *B. indicus* circa 4,000–2,000 years BP from their putative centers of domestication (*Meghen, MacHugh & Bradley, 1994*; *Freeman et al., 2004*). However, the zebu cattle were also reintroduced to the African continent by Arab traders around ~699–640 years AD following the death of the Prophet (*Bradley et al., 1998*). Penetration of the predominant African taurine to West Africa was amid 4,000 years BP (*Marshall & Hildebrand, 2002*). The post-introduction of the zebu cattle led to their spread sporadically to West Africa circa 1,400 years ago (*Meghen, MacHugh & Bradley, 1994*; *Hanotte et al., 2002*) from East Africa hypothesized to be the original entry point of zebus in Africa (*Gifford-Gonzalez & Hanotte, 2011*). To date, the origin and migration of ancient and modern African cattle is still under strong debate due to conflicting archaeological and genetic evidence (*Hanotte et al., 2000*, *2002*; *Magnavita, 2006*), therefore, unravelling of any possible clues is still at large and with very crucial pressure sought-after by scientist and archaeologist.

In the past decade scientist have been using mitochondrial DNA (mtDNA) markers in the control region and the entire mtDNA genome to understand phylogenetic tree models for describing origin, divergence times and domestications of cattle and their population expansion to different parts of the globe (*Achilli et al., 2008*; *Ho et al., 2008*; *Horsburgh et al., 2013*; *Olivieri et al., 2015*). MtDNA studies on cattle have grouped them in two major lineages represented by haplogroup T for *B. taurus* (*Troy et al., 2001*) and haplogroups I1 and I2 (*Chen et al., 2010*) both defining the genetic lineage of *B. indicus*. Within the haplogroup T, sister sub-haplogroups T1, T2, T3, T4 and T5 were revealed (*Mannen et al., 2004*; *Achilli et al., 2008*, *2009*). The haplotypes T2 and T3 are the dominant haplotypes in the Middle East and Europe and also in Africa at very low frequency (*Beja-Pereira et al., 2006*; *Olivieri et al., 2015*). Majority of the mtDNA haplotypes in African cattle are defined by T1 haplogroup (*Troy et al., 2001*; *Horsburgh et al., 2013*), with no evidence of zebu mtDNA haplotypes been reported so far. There has been some more discoveries of other haplotypes P, Q, R, E and C especially when analysis of whole mtDNA genome was applied (*Achilli et al., 2008*, *2009*; *Bonfiglio et al., 2010*), with only haplotype Q (Q1) being reported recently in the African cattle (*Olivieri et al., 2015*).

Genetic studies on paternal Y-chromosomal DNA markers have previously reported the frequency of both haplogroups Y1 and Y2 for taurines and of Y3 haplogroup an

exclusive for zebus (*Götherström et al., 2005*; *Bonfiglio et al., 2012a*; *Ginja et al., 2010*). Some authors like *Perez-Pardal et al. (2010)*, *Álvarez et al. (2017)* and *Chen et al. (2018)* have reported both Y1 and Y2 haplotype frequencies in sub-Saharan and Mediterranean regions in modern African cattle at considerable varying frequencies. However, majority of the African cattle have been reported to be more of zebu background (*Hanotte et al., 2000*) solely defined by haplogroup Y3 and are distributed at much higher frequency than Y1 or Y2, enormously across Africa encompassing both the North, West, East, Central and southern African regions (*Perez-Pardal et al., 2010*, *2018*; *Álvarez et al., 2017*; *Ginja et al., 2019*).

Nigeria, a country in West Africa harbors both the taurine and zebu cattle (*Rege, 1999*) and their crossbreds (*Loftus et al., 1994a*). Apart from their economic benefit such as meat, milk, and skin, the Nigerian cattle are observed as important idols in ceremonial rituals and also utilized in drafting and ploughing during farming. Acquisition of the current knowledge regarding the genetic status of Nigerian cattle is crucial for conservation and utilization of their genetic resources. Previous genetic diversity studies in West African cattle, particularly in Nigeria are limited by the few number of populations (*Bradley et al., 1994*; *Loftus et al., 1994a*, *1994b*; *Bradley et al., 1996*; *Perez-Pardal et al., 2018*). This could imply that the actual extent of the genetic diversity of Nigerian cattle still remains an enigma.

To disclose the genetic diversity in Nigerian cattle, we employed the use of both mtDNA and Y-chromosomal markers, which have been widely used in assessing the diversity and phylogeographic structure of many domestic animals (*Lindgren et al., 2004*; *Ramirez et al., 2009*; *Wang et al., 2014*; *Álvarez et al., 2017*). In this study, we evaluated variation in the mtDNA D-loop and Y-chromosome using 139 Nigerian cattle samples. Due to the nature of husbandry management in most of African countries, the assessment of genetic variation in Nigerian cattle was conducted based on their sampled locations such as from North West (Zamfara, Kano, Katsina, Kaduna and Sokoto States), North East/central (Taraba and Plateau States) and western (Oyo State) regions of Nigeria (Fig. 1A).

## MATERIALS AND METHODS

### Ethical considerations

All experimental procedures in the present study were performed in accordance to Research Guidelines for the Institutional Review Board of Kunming Institute of Zoology, Chinese Academy of Sciences (SMKX2017009) and the Central Abattoir, Ibadan, Ministry of Agriculture and Rural Development, Oyo State, Nigeria. Several cattle are usually transported from farms in different states in Nigeria to the Central Abattoir in Ibadan. Therefore, all cattle samples were then collected at the Central Abattoir, Ibadan.

### Sampling and data collection

A total of 139 Nigerian cattle individuals (119 females and 20 males) were sampled from farmer's herds in eight different States in Nigeria as follows (Fig. 1A); Kaduna State ($n = 19$ females; $n = 3$ males), Kano State ($n = 4$ females; $n = 2$ males), Katsina State ($n = 4$ females; $n = 2$ males), Sokoto State ($n = 27$ females; $n = 4$ males), Mambilla plateau in Taraba
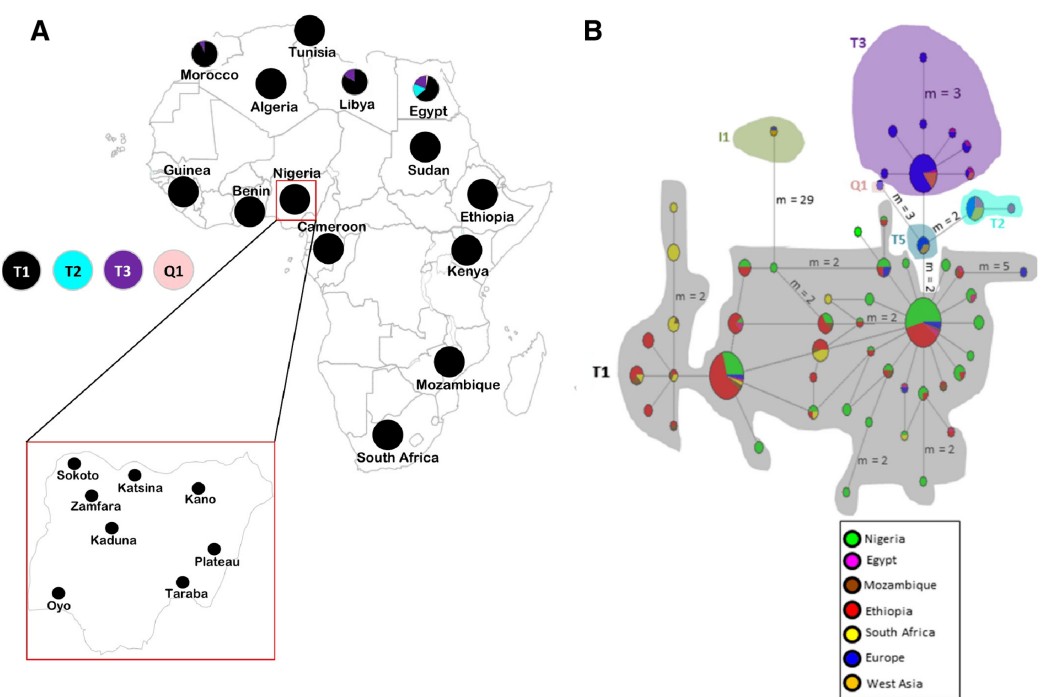

**Figure 1 Sampling locations of cattle in Nigeria and the network of 420 cattle samples based on 636 bp of the mtDNA D-loop region.** (A) Map of cattle sampling locations in Nigeria (Zamfara, Kano, Katsina, Kaduna and Sokoto from North West; Taraba and Plateau from North East/Central; and Oyo from the West) and the haplogroup distribution based on mtDNA across Africa . The maps of Africa and Nigeria were generated by an online version of the SmartDraw 2012 software (https://cloud.smartdraw.com/editor.aspx?templateId=3a99cb96-00dd-4767-bca6-61a59bd9ad60&flags=128). (B) Median-joining network of 420 cattle samples constructed by using NETWORK v 4.6 ( *Bandelt, Forster & Rohl, 1999*). Reference sequences used for haplotype network analysis included: Europe, n = 76 (*Loftus et al., 1994a*; *Lai et al., 2006*; *Achilli et al., 2008*; *Hiendleder, Lewalski & Janke, 2008*; *Bonfiglio et al., 2012b* and AF034438–AF034446 were retrieved from the GenBank); West Asia, n = 16 (*Achilli et al., 2008*); Egypt, n = 31 (*Bonfiglio et al., 2012b*; *Olivieri et al., 2015*); Ethiopia, n = 126 (*Dadi et al., 2009*; *Bonfiglio et al., 2012b*); Mozambique, n = 16 (JQ684029–JQ684045 were retrieved from the GenBank), South Africa, n = 34 (*Horsburgh et al., 2013*) and two additional Nigerian samples mined from GenBank (Accession no. L27731 and L27730). Sizes of the circles are proportional to haplotype frequencies. m, refers to number of mutation steps and those not indicated are just one step mutation. Colours indicate the geographical distribution of the sampling locations across Africa, Europe and West Asia as shown by the legend in (B).

State (n = 35 females; n = 2 males), Zamfara State (n = 8 females; n = 2 males), Oyo State (n = 1 female; n = 2 males) and Jos city in Plateau State (n = 21 females; n = 2 males). During sample collection, genetically unrelated cattle from the eight states in Nigeria, ecological and geographical perspectives were considered. This included randomly sampling of at least two animals per household and only from those households located approximately 0.5 km further apart. Farmers were also interviewed on the pedigree information of their animals prior to carrying out blood collection. Blood samples were kept in 95% ethanol at room temperature before transportation to the laboratory. Samples were stored at 4 °C for immediate use, or at −80 °C for later use.

Yak (*Bos grunniens*: Accession no. MN398192, *Huang et al., 2020*), was used as an outgroup in phylogenetic tree analysis. A total of 420 sequences based on 636 bp (of the

D-loop region) that included 119 individuals from Nigeria, 76 from Europe, 16 from West Asia and 209 from other African countries (31 Egyptian cattle, 16 Mozambiquan cattle, 126 Ethiopian cattle, 34 Nguni cattle from South Africa, and two additional samples from Nigeria) were used in mtDNA analyses (more information in Fig. 1 legend part b; Table S1A). For Y-chromosome analyses only few representative samples of the three major haplogroups of cattle were used as detailed in Table S2.

## DNA extraction, PCR and sequencing

Genomic DNA was extracted from ~5 ml of blood following phenol-chloroform method (*Sambrook & Russell, 2001*). We amplified 636-base pair of the D-loop region of mtDNA, using primers constructed from L27712 D-loop sequence for both forward and reverse primers (Table S3) (*Loftus et al., 1994a*). The mtDNA amplification and sequencing reactions were carried out in a total of 25 μl PCR reaction mixture using ~40 ng of mtDNA, 10 pmol of each primer, 2.5 mM dNTPs and 5 units of Takara Taq DNA polymerase in a 10 pmol reaction buffer containing 1.5 mM $MgCl_2$. Amplifications were carried out in a thermocycler for at least 35 cycles as follows: 95 °C for 5 min, 94 °C for 45 s, 58 °C for 30 s, 72 °C for 1 min 30 s and a final extension of 7 min at 72 °C. The quality and confirmation check were performed using 2% agarose gel and visualization under UV transilluminator.

We have sequenced 286 bp of the bovine Y-chromosome in 20 Nigerian cattle (Tables S2 and S3) in order to identify polymorphic sites in the X-degenerate male specific regions of the bovine Y-chromosome (MSY) (*Chang et al., 2013*). In particular, a region of the zinc finger protein Y-linked (ZFY) gene similar to that of *Homo sapiens* (human) intron 10 was used as previously described (*Ginja, Telo da Gama & Penedo, 2009*; Table S3) for the amplification and sequencing of all 20 samples at the length of 286 bp. We applied the same PCR conditions used in the amplification of mtDNA with exceptional of the annealing temperature of 53 °C. The quality control checks as described in mtDNA were also carried out on all the 20 PCR amplicons. The amplified mtDNA and Y-chromosomal DNA fragments were subjected to sequencing procedures such as purification of the PCR products with Exo - SAP-IT Cleanup kit as per manufacturer's instructions (Affymetrix). Subsequently, sequencing reactions were carried out using the BigDye[TM] Terminator v3.1 Cycle Sequence Kit (Applied Biosystems, Foster City, CA, USA) and the generated products from this step were further purified by alcohol precipitation. Lastly, we used ABI PRISM 3730 automated DNA sequencer (Applied Biosystems, Foster City, CA, USA) to sequence the purified products. All sequences were assembled in SeqMan Lasergene package in DNASTAR software.

## Data analysis
### Sequences check and alignment

The assembled DNA sequences of 139 individuals were exported into MEGA X ver 10.1.7 software (*Kumar et al., 2018*) for alignment with other cattle populations mined from Genbank for both mtDNA D-loop (Tables S1A and S1B) and ZFY Y-chromosomal markers (Table S2). Multiple sequence alignments of the D-loop region and ZFY gene of

the Y-chromosome were carried out using CLUSTAL W package (*Thompson, Higgins & Gibson, 1994*) integrated in the MEGA software. All sites containing alignment gaps were excluded from the analysis. Our data involved amplification and sequencing of the 636 -base pair (bp) mtDNA D-loop region and 286 bp ZFY gene of 119 and 20 Nigerian cattle samples respectively. Variations in the D-loop region were detected by assembling all forward and reverse sequences against the reference *B. taurus* mtDNA sequence ((GenBank accession no. V00654); (*Anderson et al., 1982*)). Variations in the Y-chromosome of the X degenerate region located at intron 10 of ZFY gene were also determined. The alignments and assembly of the Y-chromosome sequences were carried out similarly following the protocol used by *Ginja, Telo da Gama & Penedo (2009)* where sequences from distantly related species were used as described by *Götherström et al. (2005)* but using *Bison bison*, *Bos frontalis*, and *B. grunniens* sequences (*Verkaar et al., 2004*). The polymorphic sites in the Y-chromosome male specific region were identified using a Y-chromosome *B. taurus* reference genome (GenBank Accession no. AF241271; *Lawson & Hewitt, 2002*).

## Genetic diversity and haplogroup classification

MitoToolPy_Linux (*Peng et al., 2015*) was used to determine haplogroup distribution across Africa by analyzing a 240 bp fragment length of the D-loop region of mtDNA involving 609 cattle samples in total (including the 119 Nigerian cattle sequenced in this study (Fig. 1A; Table S1B)); (reference data were retrieved from: *Loftus et al., 1994a*; *Bradley et al., 1996*; *Troy et al., 2001*; *Beja-Pereira et al., 2006*; *Dadi et al., 2009*; *Bonfiglio et al., 2012b*; *Horsburgh et al., 2013*; *Olivieri et al., 2015*). We used DnaSP v5 (*Librado & Rozas, 2009*) to determine the haplotypes in 420 mtDNA cattle sequences and the assignment of bovine Y-chromosome haplogroups for Nigerian cattle. Genetic diversity, was assessed by using Arlequin v3.5 (*Excoffier & Lischer, 2010*) and expressed in terms of the total number of haplotypes ($H$) and polymorphic sites (PS), haplotype diversity (HD), nucleotide diversity ($\pi$), the mean number of nucleotide differences (Df) and their standard deviations (SD) estimated across all African populations used in this study. Notably, the comparisons of the genetic diversity estimates were considered for only those populations with sample size above 5.

## Phylogenetic tree analyses

To investigate the evolutionary relationship of Nigerian cattle with other cattle samples mined from GenBank (Tables S1A and S2), the same version of MEGA software was used to construct a rooted neighbor-joining (NJ) phylogenetic tree (*Saitou & Nei, 1987*) using the Maximum Composite Likelihood evolutionary distance approach (*Tamura, Nei & Kumar, 2004*) and bootstrap test was employed at 1,000 replications so as to assess the confidence of each node (*Felsenstein, 1985*). To further visualize the genetic relationships between the haplotypes and identifying the number of unique mtDNA D-loop haplogroups present in the 420 dataset, the median-joining (MJ) network (*Bandelt, Forster & Rohl, 1999*) was generated by using the default setting weights of both

transversions and transitions as implemented in Network v4.6 software (www.fluxus-engineering.com).

## Population genetic structure and demographic dynamic profiles

To infer the matrilineal genetic variation within populations, among populations, and groups of populations, analysis of molecular variance (AMOVA) was carried out following 50,000 permutations in Arlequin v3.5 software. The analysis was conducted for Nigerian cattle at various hierarchical levels *viz* the Nigerian cattle as a single cluster, Nigerian cattle vs the other African countries but also vs cattle from Europe and West Asia. The levels of significance in each hierarchical cluster tested were evaluated using $F_{ST}$ parameter at a significant *P* level of 0.05.

To investigate the population dynamics and demographic patterns of Nigerian cattle population, mismatch distribution patterns were estimated (*Rogers & Harpending, 1992*) with respect to their geographical regions for North West and North East. The chi-square test of goodness of fit and Harpending's raggedness index "*r*" (*Harpending, 1994*) statistics were also calculated to assert the significance of the deviations of the sum of squares differences (SSD) observed from the simulated model of demographic expansions determined by 1,000 coalescent simulations. Demographic statistical parameters for Tajima's *D* (*Tajima, 1989*) and Fu's $F_S$ (*Fu, 1997*) were also estimated by using Arlequin v3.5 software to further complement the mismatch distribution patterns.

## RESULTS

### MtDNA genetic diversity

In this study we evaluated variations in the mtDNA D-loop of 119 Nigerian cattle together with 301 global cattle sequences based on 636 bp from Egypt, Ethiopia, Mozambique, South Africa, Europe and West Asia available in the GenBank. The sequences generated in this study have been deposited in the GenBank with accession numbers MT362777–MT362895. There were 153 variable sites scored in all 420 cattle samples that defined 275 haplotypes (Table S4) and 80 of them assigned to Nigerian cattle sequenced in this study (Table 1). Most of the Nigerian cattle in the current study possess unique haplotypes (80%) and the remaining ones were shared with other African and European cattle. The lowest level of haplotype diversity (0.983 ± 0.009) was observed in cattle from North East while the highest (0.984 ± 0.007) was observed in the North Western region. Estimated haplotype diversity (HD) across all Nigerian individuals was 0.985 + 0.005 (Table 1). This value observed is lower than the haplotype diversity of Egyptian and Mozambican cattle populations but was higher compared to Ethiopian and South African cattle.

### Haplogroup classification and phylogenetic trees using mtDNA

The haplogroup distribution across Africa shows the majority of African cattle are of *B. taurus* T1 the widely known *B. taurus* haplogroup for African cattle (Fig. 1A; Table S1B). The Nigerian cattle in this study were all classified into haplogroup T1 (Tables S1A and S1B) and majority of them constitute of haplogroup T1a similar to

**Table 1 Genetic diversity of cattle in Africa based on mtDNA D-Loop.**

| Population | $N$ | PS | $H$ | HD (SD) | $\pi$ (SD) | Df |
|---|---|---|---|---|---|---|
| 1. Nigeria | 119 | 67 | 80 | 0.985 (0.005) | 0.051 (0.029) | 3.403 |
| (a) North West[1] | 62 | 44 | 44 | 0.984 (0.007) | 0.076 (0.044) | 3.338 |
| (b) North East/Central[2] | 56 | 52 | 43 | 0.983 (0.009) | 0.067 (0.038) | 3.462 |
| (c) West[3] | 3* | 4 | 3 | 1.000 (0.272) | 0.667 (0.598) | 2.667 |
| 2. Egypt | 31 | 46 | 30 | 1.000 (0.082) | 0.010 (0.005) | 6.391 |
| 3. Mozambique | 16 | 18 | 15 | 0.992 (0.025) | 0.180 (0.109) | 3.233 |
| 4. Ethiopia | 126 | 70 | 83 | 0.969 (0.009) | 0.005 (0.003) | 3.365 |
| 5. South Africa | 34 | 17 | 24 | 0.961 (0.019) | 0.006 (0.003) | 3.606 |

**Notes:**
[1] Cattle samples from Zamfara, Kano, Katsina, Kaduna and Sokoto States.
[2] Samples from Taraba and Jos, Plateau States.
[3] Samples from Ibadan, Oyo State.
* Two additional downloaded samples from GenBank (Accession no. L27731 and L27730). The estimation of haplotype and nucleotide diversity based on 636 bp mtDNA D-loop sequence was carried out by ARLEQUIN v. 3.5 (*Excoffier & Lischer, 2010*) software.
$N$, sample size; PS, the number of polymorphic sites; $H$, the number of haplotypes; HD, haplotype diversity; $\pi$, nucleotide diversity; Df, the mean number of nucleotide differences and SD, standard deviations.

findings by *Olivieri et al. (2015)*. Apart from Nigeria, other African countries in the Northern part of Africa particularly Egypt, Libya and Morocco also carried other types of *B. taurus* haplogroups T2, T3 and Q1. No traces of *maternal* lineages were observed in Nigerian cattle samples sequenced in this study.

To obtain further insights into the haplotype relationships, the network analysis (Fig. 1B) and phylogenetic tree (Fig. S1) were constructed using the 636 bp sequences of 119 Nigerian cattle and 301 other sequences retrieved from the GeneBank. The phylogenetic tree depicted two major lineages of cattle, *B. taurus* (T1, T2, T3, T5 and Q1) and *B. indicus* (I1 and I2) lineages as expected (*Chen et al., 2010*; *Achilli et al., 2008*). All Nigerian cattle have been placed together with the rest of taurine cattle individuals separately from the zebu cattle. The MJ network depicted similar pattern where haplogroup I1 for zebu lineage was separated from all the taurine haplogroups. Furthermore, the network revealed two major clustering: the first cluster showed grouping of Nigerian cattle with African and European cattle; while the second cluster did not contain Nigerian individuals. Cattle samples in the first cluster exhibit a star-like pattern, a matrilineal characteristic which signifies a signature of population expansion. Furthermore, we found two Nigerian individuals which showed interesting results. These individuals have been placed within the monophyletic clade containing individuals of haplogroups T1, T5 and Q1 from Europe, which could indicate a similar pattern of origin with European cattle (*Achilli et al., 2008*). One of the individuals (71_TAR_NIG) possess similar mutations as T5 (*Achilli et al., 2008*) at g.16255 and 16197 (Table S5) which concurs with the phylogenetic tree results. The second individual (111_KAD_NIG) appears like a basal lineage to this monophyletic clade where individuals of haplogroups T5, T1 and Q1 are clustering. Our analysis also shows some shared polymorphism between Nigerian cattle (71_TAR_NIG and 111_KAD_NIG), the *Bison bison* and *B. grunniens* at

transitional mutations G/A: g.15921 and T/C: g.16204 (Table S5) as previously detected by *Achilli et al. (2008)*.

## Population genetic structure and historical demographic dynamics

Analysis of molecular variance incorporating the eight populations from Nigeria showed that more than 99% of the total genetic variation present in Nigerian cattle occurred within individuals (Table S6). Furthermore, when analyzing the genetic variation between Nigerian cattle and other cattle populations AMOVA showed that, 62 to 93% of the total variation between Nigerian cattle and other African cattle populations occurs within individuals, with the highest variation observed between Nigerian and Mozambican cattle (93.46%). Generally, all attempts to explore the differentiation between Nigeria and other populations were attributable to within-population variance ($P < 0.05$) with exceptional of variation among populations within groups for Nigerian and European cattle ($F = 0.07375$, $P = 0.00004$). The $F_{ST}$ distance values between most Nigerian cattle sub-populations were low (Table S7). However, it was considerably high between Nigerian cattle and other African cattle populations showing significant genetic variation between them at $P < 0.05$ with exceptional for cattle populations between Ethiopia and Mozambique ($F_{ST} = -0.00548$, $P = 0.57129$). Our AMOVA results complemented by genetic distance estimates suggest general absence of maternal genetic structuring in Nigerian cattle sub-populations, likely due to extensive genetic intermixing within the country.

To elucidate the demographic dynamics of Nigerian cattle, mismatch distribution patterns, for each geographical region in Nigeria were assessed (Fig. S2). The mismatch distribution patterns were unimodal, however the pattern deviated significantly from expected under a null hypothesis model of either spatial or demographic expansion due to significant values obtained for Sum of Squared deviation (SSD) and Harpending's Raggedness index (HRI) (Table S8). The significant values for SSD and HRI indicate a bad goodness of fit test, which does not support the scenario of population expansion. The values for Tajima's $D$ $-2.315$ (P < 0.05) and Fu's $F_S$ statistics $-26.203$ (P < 0.05) on the other hand were both negative and significant indicating an agreement of recent population growth and expansion respectively. Cattle population from the Western region of Nigeria showed Tajima's $D$ value of 0 possibly due to only three individuals were sampled, which might indicate that the population in that region evolved as per mutation drift equilibrium with no evidence of selection.

## Genetic polymorphism of Y-chromosome and haplogroup distribution

In Fig. S3, we provide detailed information on haplogroup distribution based on Y-chromosomal markers in Africa (*Perez-Pardal et al., 2018*; *Ginja, Telo da Gama & Penedo, 2009*). The sequences generated in this study have been deposited in the GenBank with accession numbers MW284951–MW284970. Our results show that all Nigerian cattle belong to haplogroup Y3 solely a *B. indicus* haplogroup due to similar mutations (Table 2; Supplemental Data S1). Notably, in addition to previously reported mutations that further classify zebu cattle into haplotype Y3 families as pointed out by *Chen et al. (2018)* and

**Table 2 Mutations that describe the distinction of the three major haplogroups of cattle based on ZFY intron 10 gene. The two genomic sites that separate each distinct haplogroup are in bold. The seven mutations observed in Nigerian cattle are italicized.**

**ZFY intron 10**

| Haplogroup | Reference | Alternative | Publication |
|---|---|---|---|
| Y1 | C | C | |
| | GT | — | |
| Y2 | C | **C** | *Götherström et al. (2005)* |
| | GT | **GT** | *Ginja, Telo da Gama & Penedo, 2009* |
| Y3 | C | **T** | |
| | GT | GT | |
| Y3 | C | T | |
| | GT | GT | |
| g.797 | *A* | *G* | |
| g.786 | *A* | *T* | |
| g.800, 804 | *A* | *C* | This study |
| g.784, 803 | *T* | *A* | |
| g.802, 805 | *T* | *G* | |
| g.802, 803, 804 | *T* | *C* | |
| g.801 | *C* | *T* | |

*Perez-Pardal et al. (2018)*, we also show new mutations in Nigerian cattle possibly not previously reported. These include those SNPs observed between g.784 and g.805 bp (Table 2; Supplemental Data S1). Multiple sequence alignment with haplotypes defining the three major haplogroups of cattle (Y1, Y2 and Y3) (*Nijman et al., 2008*; *Ginja, Telo da Gama & Penedo, 2009* and Table S2) revealed the existence of seven mutations including mutations A > G, T > C and T > G that distinguish Y3a (or $Y_C$) and Y3b (or $Y3_A$, $Y3_B$) haplotype families (Table 2) grouping Nigerian cattle into zebu Y3b sub-haplotype family (Supplemental Data S1) due to having similar polymorphic information at sites A/A and T/T (Supplemental Data S1). Majority of the Nigerian cattle shared haplotype Hap_7Y3 together with cattle from Asia (Fig. 2) which may signify possible shared origin from the most recent common ancestor (TMRCA). The phylogenetic tree shows that Nigerian cattle cluster separately from European Y1 or Y2 haplogroups (Fig. S4). The phylogenetic tree showed some of the Nigerian cattle clustering together with *B. grunniens* (yak) and *B. bison*, these two latter populations are considered distantly related populations to the major lineages of domestic cattle (Fig. S4).

The overall genetic diversity for Nigerian cattle measured by the haplotype diversity and the mean number of pairwise difference is 1.000 ± 0.016 and 1.679 ± 1.027 respectively (Table S9). The genetic diversity detected in some African cattle populations/breeds as per *Perez-Pardal et al. (2018)* and *Ginja et al. (2019)* although different Y-chromosomal markers such as microsatellite markers were used, the estimates still provide reliable information needed for comparison with our samples (Table S9).

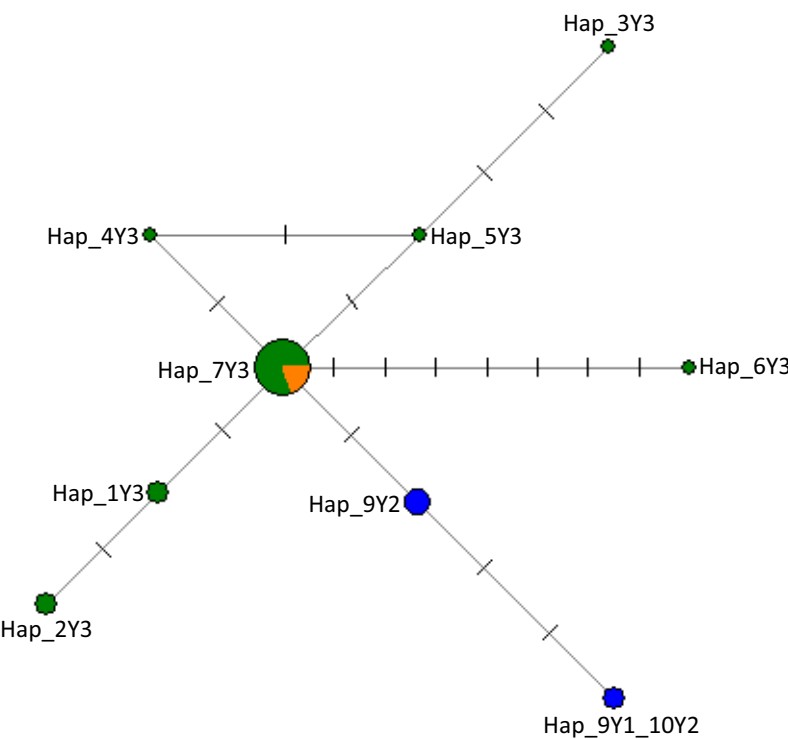

**Figure 2** **The network of 28 cattle samples based on 286 bp of the ZFY Y-chromosome region.** Sizes of the circles are proportional to haplotype frequencies and the number of mutation steps in each branch are given. Colours indicate the geographical distribution of the samples as follows: dark green, Nigeria; orange, Asia; and blue, Europe. The source of the data for the Y haplogroups was retrieved from *Nijman et al. (2008)*, *Ginja, Telo da Gama & Penedo (2009)*, MF683853, MF683854 were retrieved from the GenBank.

## DISCUSSION

In this study we examined the genetic variation across eight Nigerian cattle populations, which represent samples from one country in West Africa. The results based on mtDNA D-loop revealed 80 haplotypes from 119 Nigerian cattle sequences, which showed a haplotype diversity of 0.985 ± 0.005. The haplotype diversity in Nigerian cattle is lower compared to those of Egyptian cattle $H = 1.0 \pm 0.008$ (Olivieri et al., 2015). These findings signify a high level of maternal genetic variation in Nigerian cattle. AMOVA on the other hand suggests a general absence of maternal genetic structuring in Nigerian cattle that may have been as a result of extensive genetic intermixing within the country. Studies show that genetic intermixing is caused by rigorous transportation of domestic animals like goats, cattle, and sheep from place to place as a result of their valuable resource for economic trade or cultural exchange (*Tarekegn et al., 2018*). Notably, recent study has revealed the impact of African cattle pastoralism in admixture of cattle across the continent, and this admixture or intermixing has been well observed in the African humped cattle (*Kim et al., 2020*). It is generally known that most of the genetic variation in cattle elucidated by matrilineal genetic information is attributable to geographical differences (*Hanotte et al., 2002*), which can be sourced at the regional or continental levels (*Dadi et al., 2009*) rather than by their morphological disparities or differences in their

origins (*Álvarez et al., 2017*). We observed this scenario in our study samples where the variation was explained only within populations from different geographical regions in Africa such as between pairs of Nigerian cattle with other African cattle populations (Table S6). On the other hand, although majority of variation can be explained within a population at individual level, the variation has always been nonsignificant reflecting lack of genetic structuring. This trend of unstructured matrilineal populations within Africa has been reported consistently throughout the entire continent (*Álvarez et al., 2017*). One of the possible reasons could have been due to the influx of a continual zebu introgression especially following the great rinderpest disease that wiped out nearly 5.2 million head of cattle (*Hanotte et al., 2000*, *2002*). Nonetheless, the hybridization process between zebu and taurine which is practiced throughout the continent, might somewhat explain this unstructured scenario. This is usually attributed by the need of farmers to have a breed of cattle that can have both of the genetic attribute including resistance to disease such as trypanosomiasis and at the same time to be capable to withstand adverse tropical environmental conditions such as drought or hot or humid climatic environments (*Kim et al., 2017*).

We also observed similar genetic diversity pattern based on Y-chromosome analysis where the haplotype diversity detected was higher compared to that observed in a previous study (*Perez-Pardal et al., 2018*) for Nigerian cattle possibly due to a single breed population sampled, albeit it was similar to haplotype diversity of cattle from India and Central Asia (*Perez-Pardal et al., 2018*). Generally, in comparison with other cattle populations in Africa, our study indicated the highest genetic diversity of $1.000 \pm 0.016$ with the lowest in Zebu_Peul breed population from Burkina Faso (Table S9). Nigerian male cattle samples have been observed with high genetic diversity extrapolated by Y-chromosomal single nucleotide polymorphic marker which is consistent with the scenario of the unique zebu alleles mostly found only in West Africa (*Perez-Pardal et al., 2010*, *2018*). This was hypothesized by these authors that zebus in West Africa have got unique genetic background probably because of genetic contribution from local ancient humped cattle. However, even though recent zebu introgression could have occurred into this region, may have not contributed much to the genetic affinity with other zebus from elsewhere possibly because the introgressions were female mediated (*Hanotte et al., 2000*) which can also be described in the case of Kuri cattle samples of Lake Chad (*Meghen et al., 2000*) or perhaps the restocking of African zebu sires from East to West during the early 20th century was relatively low (*Álvarez et al., 2017*). The high genetic divergence between zebu of West Africa and those from Asia was recently unveiled by *Perez-Pardal et al. (2018)* which further strengthens this speculative hypothesis.

The present-day zebu-like cattle in West Africa were products of crossbreeding events between the West African taurine cattle such as N'Dama and imported zebu cattle directly from Asia (*MacHugh et al., 1997*; *Kim et al., 2017*) or rather indirectly with zebu from East Africa. We computed the genetic relationships among Nigerian cattle by using MJ network (Fig. 1B) and NJ tree (Fig. S1). We observed all Nigerian cattle to be exclusively of taurine T1, an African specific haplogroup, and morphological resemblance of *B. indicus* such as the presence of the humps. This scenario is in accordance with similar

observations by *Loftus et al. (1994a)* but also a similar case was depicted in zebu breeds of the Americas where they were typically found with mitochondrial taurine haplogroup in their genomes (*Ginja et al., 2019*). These observations further explain that zebu introgression is typically male-mediated and that this scenario took place in many places around the globe. In the light of this argument with respect to Nigerian cattle in West Africa, most farmers in this region preferred zebu of male lineage due to their massive muscle size but also being resistant to rinderpest that taurines are not, even though them being less vigorous towards trypanosomiasis, a disease prevalent in tsetse regions of both West and Central Africa whereby taurine cattle depict resistance to the disease (*Grigson, 1991*; *Ibeagha-Awemu et al., 2004*) The complete absence of Asian zebu mtDNA in Nigerian cattle samples suggests that crossbreeding events were mainly through the imported Asian male zebus (*Loftus et al., 1994a*, *1994b*; *Bradley et al., 1996*). Nonetheless, this scenario could have been attributed by the continued adoption of trypanotolerant breeds that probably led to total loss of the *B. indicus* mtDNA lineages or its vulnerability towards population bottleneck. This has also been observed elsewhere by several studies particularly cattle from North-East Asia (*Mannen et al., 2004*). *Bradley et al. (1994)* and *Perez-Pardal et al. (2018)* further stress this observation extrapolating the importation of probably only male zebus into West African region. Other factors such as droughts and the great rinderpest disease outbreak or unbiased selection of cattle breeds over zebu mitochondria may have contributed to the loss of any rare zebu mtDNA (*Dadi et al., 2009*).

We have also analyzed the Nigerian cattle using a single Y-chromosomal polymorphic marker in the X-degenerate region of the male specific Y-chromosome. Our findings based on mtDNA and Y-chromosome analyses conducted in this study have confirmed that Nigerian cattle are an influence of both taurine and zebu lineages from female and male genetic contributions respectively, with only zebu male specific haplotypes being detected. This illustrates a complex scenario of the genetic background in most of the present-day African cattle populations (*Álvarez et al., 2017*). Previously, mtDNA studies (*Loftus et al., 1994a*) found that African cattle are of taurine background. However, earlier studies conducted using Y-chromosomal markers (*Bradley et al., 1994*) found that African cattle had been genetically introgressed with zebu cattle from Asia. The introgressions were male mediated (*Hanotte et al., 2000*) and that majority of African cattle belong to the zebu specific haplogroup Y3. Patrilineal studies show that zebu cattle are distributed at varying frequency across the continent with more frequency observed in eastern and Central Africa but at a lower frequency in the western and southern parts of Africa (Fig. S3). This decrease in frequency of zebu haplogroup Y3 as one moves towards West and South of Africa is possibly due to the presence of *B. taurus* European haplogroups Y1 and Y2 in these regions (Fig. S3). A study by *Decker et al. (2014)* had previously demonstrated similar observations using autosomal single nucleotide polymorphisms (SNPs) markers. This unevenly distribution of zebu allele and introgression in the African continent is possibly attributed to selection of breeds by farmers based on disease tolerance such as tolerance to trypanosomiasis or ability to withstand adverse environmental conditions such as drought (*Hanotte et al., 2000*). Although, some of the previous studies conducted in West African cattle detected

European Y1/Y2 haplogroups (*Perez-Pardal et al., 2010*; *Ginja et al., 2019*), it is surprisingly enough that our study samples from Nigeria did not yield any of these, but rather showed that all Nigerian cattle are of zebu Y3 origin. We believe that, the undetected Y1 or Y2 haplotypes in Nigerian cattle might possibly indicate low or no influence of European cattle in these studied samples, coinciding with what was reported previously by *Hanotte et al. (2000)* of 0% Y- chromosomal taurine alleles in Nigerian cattle. Nonetheless, it's probable that our sampling coverage did not cover enough samples from western part of Nigeria where most cattle of haplotype Y1 and Y2 are likely to be found. Similarly, this was shown in phylogenetic tree (Fig. S4), where Nigerian cattle somehow clustered separately from European Y1/Y2 haplotypes. Some studies based on additional Y-chromosomal markers, including STRs, have further indicated that *B. indicus* haplogroup Y3 is composed of three sub-haplotype families which include $Y3_A$, $Y3_B$ (also defined as Y3b nomenclature), and $Y3_C$ (or Y3a) (*Chen et al., 2018*; *Perez-Pardal et al., 2018*), with $Y3_A$ being a cosmopolitan haplotype, while $Y3_B$ is exclusively found in West African cattle. These three haplotypic zebu families co-existed together with their counterpart taurine in approximately 200,000 years ago (*Loftus et al., 1994a*; *Ho et al., 2008*; *Murray et al., 2010*). However, they diverged at different times among themselves before the start of domestication, with the most recent divergence occurring between West and East African zebus (*Perez-Pardal et al., 2018*).

Moreover, our study observed that the Nigerian cattle of Y3 haplogroup can be further classified into sub-haplotype family Y3b based on the nomenclature adopted by *Chen et al. (2018)* (Supplemental Data S1) with several zebu strains (Fig. 2) which possibly reflect less intensive selection in African countries as observed in Nigeria compared to other regions for instance Europe (*Xu et al., 2015*). Our findings indicate a possibility of gene flow or backcrossing of cattle with wild stock from other related bovine species which may have had occurred probably during initial stages of early domestication in the Eurasia prior to population expansion or at later instances post domestication around 2,000 years ago (*Chen et al., 2010*; *Perez-Pardal et al., 2018*). These possible imprints of gene flow or introgression with wild stock or other related domestic Bovidae was inferred through the shared matrilineal mutation motifs in the wild bison (*Bison bison*), the domestic yak (*B. grunniens*) and Nigerian cattle. This was clearly confirmed by the patrilineal phylogeny where some of the Nigerian cattle clustered closely with yak, the *B. grunniens* and *B. bison* (Fig. S4). Our finding is coherent with a study by *Chen et al. (2018)*, where they were able to detect evidence of gene flow and adaptive introgression between Chinese zebu cattle and the Banteng (*B. javanicus*). Nonetheless, previous studies have also reported possibility of secondary introgression events that might have occurred from the wild oxen stock into the gene pool of domestic cattle during early domestication (*Achilli et al., 2008*, *2009*). The observed shared wild stock mutations in Nigerian cattle agrees with previous studies (*Perez-Pardal et al., 2010*, *2018*) that the $Y3_B$, also termed Y3b (by *Chen et al. (2018)*) zebu lineage in West Africa were probably intermingled with ancient humped cattle which would currently be a significant representative reservoir of male zebu biodiversity. The legacy of the possibility of subsequent introgression from local ancient wild cattle partly explains the existing large divergence between West African zebus and

zebus from Asia (*Perez-Pardal et al., 2010*, *2018*). These observations maybe speculative, and could indicate a much more complex scenario of the origin of cattle that possibly involved multiple ancestral domestication populations (*Perez-Pardal et al., 2018*) especially when considering the discrete contribution of wild ox into the gene pools of the major descendant lineages of cattle (*Murray et al., 2010*) or perhaps the remnant traces left before the divergence between domestic *Bos* species and the ancient wild ox had taken place some million years ago (*Buntjer et al., 2002*).

## CONCLUSIONS

This study reported the current genetic status and some possible new insights about the origin of cattle in West Africa using Nigerian samples from matrilineal and patrilineal perspectives. High level of maternal and paternal genetic diversity was observed in Nigerian cattle, with lack of phylogeographic structure possibly due to human mediated interventions that usually enhance severe intermixing as a result of improper husbandry management practices. The phylogenetic tree based on patrilineal analysis and the matrilineal haplogroup classification have both provided consensus evidence of a possible introgression and gene flow from wild ancient stock and other related bovine species into Nigerian cattle. We recommend carrying out in-depth population genetic studies using high-throughput technologies on complete mtDNA and autosomal genomes of Nigerian cattle in West Africa as well as the use of other Y-chromosomal markers in order to generate comprehensive genetic information on their adaptive traits, selection and demographic history.

## ACKNOWLEDGEMENTS

We appreciate all those who assisted in the study.

### Funding

This work was supported by the Sino-Africa Joint Research Center, Chinese Academy of Sciences (SAJC201611) and the Animal Branch of the Germplasm Bank of Wild Species, Chinese Academy of Sciences (the Large Research Infrastructure Funding). The Chinese Academy of Sciences President's International Fellowship Initiative provided support to Adeniyi Charles Adeola (2018FYB0003). David Heriel Mauki and Said Ismael Ng'ang'a received support from the Chinese Academy of Sciences-The World Academy of Sciences (CAS-TWAS) President's Fellowship Program for Doctoral students. The funders had no role in study design, data collection and analysis, decision to publish, or preparation of the manuscript.

### Grant Disclosures

The following grant information was disclosed by the authors:
Sino-Africa Joint Research Center, Chinese Academy of Sciences: SAJC201611.
Animal Branch of the Germplasm Bank of Wild Species, Chinese Academy of Sciences

(the Large Research Infrastructure Funding).

Chinese Academy of Sciences President's International Fellowship Initiative: 2018FYB0003.

Chinese Academy of Sciences-The World Academy of Sciences (CAS-TWAS) President's Fellowship Program.

## Competing Interests

The authors declare that they have no competing interests.

## Author Contributions

- David H. Mauki conceived and designed the experiments, performed the experiments, analyzed the data, prepared figures and/or tables, authored or reviewed drafts of the paper, and approved the final draft.
- Adeniyi C. Adeola conceived and designed the experiments, performed the experiments, analyzed the data, authored or reviewed drafts of the paper, and approved the final draft.
- Said I. Ng'ang'a performed the experiments, analyzed the data, prepared figures and/or tables, authored or reviewed drafts of the paper, and approved the final draft.
- Abdulfatai Tijjani performed the experiments, authored or reviewed drafts of the paper, and approved the final draft.
- Ibikunle Mark Akanbi performed the experiments, authored or reviewed drafts of the paper, and approved the final draft.
- Oscar J. Sanke performed the experiments, authored or reviewed drafts of the paper, and approved the final draft.
- Abdussamad M. Abdussamad performed the experiments, authored or reviewed drafts of the paper, and approved the final draft.
- Sunday C. Olaogun performed the experiments, authored or reviewed drafts of the paper, and approved the final draft.
- Jebi Ibrahim performed the experiments, authored or reviewed drafts of the paper, and approved the final draft.
- Philip M. Dawuda performed the experiments, authored or reviewed drafts of the paper, and approved the final draft.
- Godwin F. Mangbon performed the experiments, authored or reviewed drafts of the paper, and approved the final draft.
- Paul S. Gwakisa performed the experiments, authored or reviewed drafts of the paper, and approved the final draft.
- Ting-Ting Y. in performed the experiments, authored or reviewed drafts of the paper, and approved the final draft.
- Min-Sheng Peng conceived and designed the experiments, authored or reviewed drafts of the paper, and approved the final draft.
- Ya-Ping Zhang conceived and designed the experiments, authored or reviewed drafts of the paper, and approved the final draft.

## Animal Ethics

The following information was supplied relating to ethical approvals (i.e., approving body and any reference numbers):

Institutional Review Board of Kunming Institute of Zoology, Chinese Academy of Sciences (SMKX2017009) and the Central Abattoir, Ibadan, Ministry of Agriculture and Rural Development, Oyo State, Nigeria provided full approval for this research.

## Field Study Permissions

The following information was supplied relating to field study approvals (i.e., approving body and any reference numbers):

Ministry of Agriculture and Rural Development, Secretariat, Ibadan, Nigeria and Institutional Review Board of Kunming Institute of Zoology, Chinese Academy of Sciences provided field approval.

## Data Availability

D-loop data is available at GenBank: MT362777–MT362895. Y-Chromosome sequences are also available at GenBank: MW284951–MW284970.

## Supplemental Information

Supplemental information for this article can be found online at http://dx.doi.org/10.7717/peerj.10607#supplemental-information.

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
