# Peer review of "Genetic variation of Nigerian cattle inferred from maternal and paternal genetic markers"

_PeerJ, doi:10.7717/peerj.10607_

## Round 0.1 · original submission · Major Revisions

I concur with Reviewers that the manuscript requires a deep improvement in order to be acceptable.

Reviewer 1 ·

Basic reporting

This paper is relevant because it solves a gap of previous works in which samples from Nigerian cattle were scant.

However, I’ve found that the research presented has not been correctly discussed according to previous knowledge on this issue.

In my honest opinion, the subsection “demographic dynamic profiles” adds very little to the paper

Experimental design

Authors analyse a sample of Nigerian cattle using mitochondrial DNA and Y-Chromosome markers.

Authors used state-of-art methods and the information provided on sampling and results is good.

Validity of the findings

Results complements previous knowledge. However, they have not been discussed appropriately. Some issues are:

- Is lack of genetic structure in Nigerian cattle consistent with the whole African cattle scenario?

- Why didn’t you find taurine Y-Chromosomes in your samples?

- Do Nigerian cattle Y3 Chromosomes belong to the West African lineage or they result from a recent zebu introgression?

Additional comments

Manuscript peerj-50178 titled “Genetic variation of Nigerian cattle inferred from maternal and paternal genetic markers” by Mauki et al.

Authors analyse a sample of Nigerian cattle using mitochondrial DNA and Y-Chromosome markers.
The paper is relevant because it solves a gap of previous works in which samples from Nigerian cattle were scant. Authors used state-of-art methods and the information provided on sampling and results is good.

However, I’ve found that the research presented has not been correctly discussed according to previous knowledge on this issue.

1.- First of all, I was surprised when I found that the paper by Alvarez et al. (2017: Differences in genetic structure assessed using Y-chromosome and mitochondrial DNA markers do not shape the contributions to diversity in African sires. J. Anim. Breed. Genet. 134, 393-404) was not included in the list of L87-89 “To decipher the genetic diversity in Nigerian cattle, we employed the use of both mtDNA and Y-chromosomal markers, which have been widely used in assessing the diversity and genogeoraphic structure of many domestic animals”.

This paper is relevant when mtDNA and Y-Chromosome markers are used in African cattle. The well-known lack of genetic structure of African cattle at either the national or regional level occurs at the continental level as well. Differentiation between cattle groups is statistically significant for Y-chromosome markers only.

Actually, authors can use thoroughly the paper by Alvarez et al. (2017) for discussion.

2.- The paper by Alvarez et al. (2017) confirms the existence of a subfamily of taurine Y-Chromosomes in West African cattle previously reported by Pérez-Pardal et al. (2010: Y-specific microsatellites reveal an African subfamily in taurine (Bos taurus) cattle. Anim. Genet. 41, 232–241; not cited). You did not find taurine Y-Chromosomes in your samples (L275). The cause is not discussed sufficiently. May it be a simple consequence of sampling (northern Nigeria, basically)? or Can it be explained by a higher zebu introgression into Nigerian cattle (e.g. MacHugh et al., 1997 or Hanotte et al. (2000: Geographic distribution and frequency of a taurine Bos taurus and an indicine Bos indicus Y specific allele amongst sub‐Saharan African cattle breeds. Mol. Ecol. 9, 387-396; not cited)?

3.- The notation of the different Y3 subfamilies identified is confusing (L282 and related pieces of text). Please use only one notation throughout the text, explaining why. I’d suggest to use the notation by Perez-Pardal et al. (2018; cited in the text) due to the following reasons: a) this paper confirms the suggestion by Perez-Pardal et al. (2010: Multiple paternal origins of domestic cattle revealed by Y-specific interspersed multilocus microsatellites. Heredity 105, 511-519; not cited) that West African cattle have their own zebu Y-chromosome lineage; and b) this paper confirms that this African lineage, although putatively originated in India, is not present in Asia. Authors must clarify to which Y3 subfamily belong the Nigerian cattle Y-Chromosomes identified. Following Perez-Pardal et al. (2018), if belonging to subfamily Y3b they may be part of the West African “reservoir” while if belonging to subfamily Y3a they may result from a more recent zebu introgression. This will improve the Discussion section (e.g. L335-341 or L358-369).

Other concerns are the following:

i.- L236 and L322-323: don’t you think that the paper by Chen et al. (2010: Zebu cattle are an exclusive legacy of the South Asia Neolithic. Mol. Biol. Evol. 27, 1–6) would merit to be cited there?

ii.- I consider that the analyses under the title “demographic dynamic profiles” (L189, L197-205 and related pieces of text) add very little to the paper and could be removed from the text of the manuscript. However that’s an editorial decision.

iii.- I’m not a native speaker of English. The paper can be understood but I found some sentences difficult to follow (e.g. the first sentence in the Abstract section: L46-48). I would suggest to reconsider the English style of the manuscript, shortening the sentences as much as possible, and avoiding terms such as “indicine” (e.g. L343; please use “zebu”). Moreover, please use the revision of English to solve typos (e.g. L89: “genogeoraphic”).

In summary, I think that the discussion of the paper may be considerably improved including relevant previous reports on this issue. This will highlight the general interest of the research presented.

·

Basic reporting

Overall, the article is well-structured and the English used throughout the text is mostly objective and according scientific standards (see some suggestions for modifications on the annotated pdf file).

Figures and Tables are mostly relevant to the content of the article. Nonetheless, I suggest I suggest showing a network of Y-haplotypes instead of Table3 (which could be presented as supplementary material).

The introduction section is well-written and highlights the most relevant aspects related to genetic diversity and mtDNA analysis. However, in my opinion this section should also include information on mtDNA and Y-chromosomal variation found specifically in Africa (contrasting with other regions), e.g. frequency of T1 vs T3 matrilines and Y3 vs Y1/Y2 Y-lineages. The authors could also include some more recent papers regarding the genetic analysis of African cattle (this is also valid for the Discussion section, but see my specific comments in the appended pdf file).

Appropriate raw data for revision of this article was provided in the form of FASTA alignments for mtDNA and Y-chromosome sequences. But, accession numbers for public databases for the sequences derived from this study should also be provided in the respective Tables (S1, S2 and S4).

I should call attention to the fact that the ethical statement is written in Chinese language, I'm not sure if PEERJ demands for a translated copy.

Experimental design

The authors use mtDNA and Y-chromosome sequence variation to investigate the genetic composition of Nigerian local cattle. The design of the study was appropriate, as well as the methods used. The results presented here derive from original primary research that to the best of my knowledge has not been published elsewhere. The methods are well-described to allow for replication of the work done, but the cycle sequencing protocol and sequencing instrument should be mentioned (see annotated pdf file).

The statistical analyses were generally appropriate, but some aspects can be improved, in particular:
-Phylogenetic analysis of mtDNA sequences – to obtain a phylogenetic tree, the authors should use instead a method that allows for the appropriate evolutionary model to be defined as well as the corresponding parameters. This is recommended for mtDNA and should allow to recover the mtDNA phylogeny and separate major haplogroups. Regarding the mtDNA network no outgroup should be specified (see annotated pdf file).

-Population dynamics based on mtDNA sequences – I’m not exactly sure how mismatch distribution patterns can be interpreted here, perhaps a more suitable reviewer can comment on this specific analysis.

- Regarding Y-chromosomal variation, I suggest also obtaining a network of Y-haplotypes to depict genetic relationships, in particular between the Y3 haplotypes. The reduced-median algorithm may provide a better resolution and should be tested.

Validity of the findings

The findings are well-organized and presented in a logic way. There are a few things that need clarification. Regarding the mtDNA analysis, there are 2 sequences the authors classify within the T3-haplogroup. But, in my opinion, the position of these sequences in Figures 1b and S1 is not consistent with this classification. The authors should provide a justification for their decision to consider these sequences T3, while keeping in mind that other polymorphic positions may be needed to fully characterize some of the mtDNA haplogroups, namely Q and T2.

As for the Y-chromosome, the authors elaborate on possible aurochs contribution to Nigerian cattle patrilines. I find this highly speculative (there's no data for the Y-chromosome of African aurochs) and providing the polymorphisms determined for Nigeria42_Y3 are real (see my comments in the appended pdf file), most probably the reason why distinct haplotypes are found in local African cattle (often at low frequencies) is related with less intensive selection of these cattle where much of the original Y-chromosome diversity is maintained. Also, the use of artificial insemination, which is known to significantly reduce the Ne for the Y-chromosome, is probably less used in African countries in comparison to other regions such as Europe or North America. This should be discussed here.

The authors also make comparisons with the results of other studies on the Y-chromosome of African cattle, which is desirable. However, a note should be added regarding the fact that the markers used across these studies are different, including in some cases more variable Y-STRs were genotyped and used to estimate Y-diversity (for example data included in Table S9).

The discussion section can be further improved. It's a bit convoluted, where the authors use contradictory statements (see my comments in the appended pdf file). In its current form it is much more an extension of the results than a good interpretation of the cattle diversity found in Nigeria and the conservation impact of finding distinct lineages. In addition to the need for genomic studies to infer adaptive variation and demography, it also deserves a comment on the need to genotype other Y-markers to be able to detect additional genetic variation and make more reliable comparisons with other studies.

Additional comments

This study provides relevant information regarding the maternal and paternal genetic diversity found in Nigerian cattle. I made suggestions for modifications throughout the text in the annotated pdf file.

The results of this study need to be contextualized under the future perspective of analyzing other markers to characterize African cattle, e.g. autosomal whole-genome variation and other Y-chromosome markers.

Providing these minor modifications are done, it is my opinion that this study is suitable for publication.

Reviewer 3 ·

Basic reporting

This work reports an extended survey of mtDNA and Y-Chromosome markers in Nigerian cattle. It is an interesting study that certainly deserves publication but requires some revision and improvements before acceptance. Below, I outline my major concerns:
In the introductory section: some of the literature is outdated and does not report most of the recent findings regarding mtDNA and Y-chromosome patterns and origins in Africa and elsewhere. There are review articles and book chapters much more recent than some of those cited. For example, there is no reference on the most extensive work on mtDNA and Y-chromosome in African cattle (Hanotte et al 2002, Mol Ecol) and instead, it is mentioned a study on autosomal microsatellites made in the same individuals. Regarding the discussion, it is poor as there is a fair amount of literature regarding the origin and distribution of the African cattle and its archaeology, human genetics, and livestock genetics. The authors shall improve substantially this part of the MS and try to discuss their findings at the light of the recent human history in the region (Bantu expansion, European and Arab influence).
For example at L230, the authors mention that T3 haplotypes were also reported by previous works in other Northern Africa countries (although no reference is provided here), but did not venture in tracking the most probable origin of the T3 haplotypes they have found in Nigerian cattle. The available amount of data would permit the authors to go further and check whether it is related to commercial breeds such as Holstein-Friesian, Simmental, or other European breeds. Due to the tremendous lack of diversity, those cosmopolitan commercial breeds have very few mtDNA haplotypes that can be easily tracked.

Experimental design

The sample size is very different between samples and this can easily influence estimations of genetic diversity parameters.
A pitfall of this study regarding the experimental design is that although the authors mention that other studies were poor in terms of representation of Nigerian cattle, they do not consider this when they compare with the data they criticized to be poorly representative. For example, they compare the diversity between Mozambique and Nigerian Cattle (L248-252). Interestingly, they often get surprised with results that are no surprise: like most of the differences that account for most of the diversity is intra-populational and that most of the genetic structure in terms of maternal lineages in cattle, is due to intercontinental differences.
Yet another issue, is the use of an outgroup in the network analysis. What is the propose of an outgroup in such an intra-specific network? its use is confusing.
Also confusing is the nomenclature of the Y-chrom. haplotypes. The authors mention several studies but they do not say what nomenclature they have followed. They shall be more precise regarding this.

Validity of the findings

The findings are valid and worthy of publication, but the interpretation of these findings can be considerably improved.

Additional comments

The authors must improve the introductory and discussion sections. The history of West Africa has been the subject of many studies in genetic and archaeology and this can and should be used to discuss their findings on Nigerian cattle.

---

## Round 0.2 · Minor Revisions

Thank you for the resubmission.

With one exception, previous Reviewers have declined to revise the new version. I had therefore to check the suitability of the Rebuttal. It seemed to be overall satisfactory, with a few exceptions. (a) I do not accept as valid the basis for not including the Y-haplotype network (paucity of data/samples; if true, then why a Table is adequate?); (b) I do not understand why Y sequences (in contrast with mtDNA) are not deposited or provided and must be requested; (c) has a translation of ethics statement been provided?

I hope you can cope with these remaining issues in order to accept the manuscript.

Reviewer 1 ·

Basic reporting

Authors have addressed most my suggestions.
Therefore, I have no reason to delay acceptation for publication of this manuscript.

Experimental design

N/A

Validity of the findings

Their interest were previosuly assessed

Additional comments

This manuscript has been improved and I can recommend its acceptation for publication in PeerJ

---

## Round 0.3 · accepted · Accept

All requests (mine and from Reviewer 1) were satisfied.